

# PAMS-Constrained Top-Down Calibration of VOC-Speciated CMAQ Simulations

Sheng-Po Chen[*1], Chieh-Heng Wang[2], Yi-Yu Lee[3], Feng-Yi Cheng[3], Jia-Lin Wang[*4]

[1]Department of Environmental Engineering, Chung Yuan Christian University, Taoyuan, 320, Taiwan
[2]Center for Environmental Studies, National Central University, Taoyuan, 320, Taiwan.
[3]Department of Atmospheric Science, National Central University, Taoyuan, 320, Taiwan.
[4]Department of Chemistry, National Central University, Taoyuan, 320, Taiwan

*Correspondence to*: Sheng-Po Chen (g9363113@gmail.com), Jia-Lin Wang (cwang@cc.ncu.edu.tw)

**Abstract.** Accurate simulation of volatile organic compound (VOC) emissions and their role in ozone ($O_3$) formation
remains a persistent challenge in chemical transport models (CTMs). Most models rely on lumped surrogate species, limiting their ability to represent speciated VOCs and directly compare with observations. In this study, we develop an enhanced version of the Community Multiscale Air Quality model, termed CMAQ-PAMS, which explicitly incorporates 54 VOC species targeted by the Photochemical Assessment Monitoring Stations (PAMS) network in Taiwan.

We evaluate model performance during a representative high-ozone event in fall 2021 and apply a top-down calibration
approach using hourly VOC data from 12 PAMS sites. The original simulation (OrigSIM) significantly misrepresents key species, largely due to reliance on U.S.-based speciation profiles. After adjustment, the modified simulation (ModSIM) shows substantial improvements in both individual species concentrations and group-level composition (e.g., alkanes, aromatics). Notably, acetylene, a key tracer of incomplete combustion, was underestimated in OrigSIM but successfully recovered in ModSIM.

Despite accounting for only ~32% of total VOC emissions, PAMS species contribute up to 52% of modeled domestic $O_3$ formation, highlighting their disproportionate impact in VOC-limited regimes. Additionally, the CMAQ-PAMS framework enables the use of diagnostic ratios (e.g., propylene/acetylene) to identify emission sources and assess air mass aging. These findings underscore the importance of localized VOC profiling and demonstrate that the PAMS-constrained CMAQ-PAMS model provides a more chemically detailed and observationally anchored platform for ozone modeling and regulatory
applications.

## 1 Introduction

In recent years, due to the effectiveness of regulatory strategies implemented by the environmental authority (Ministry of the Environment (MOENV), Taiwan), most primary pollutants have shown a declining trend (Chen et al., 2021). However, ozone ($O_3$) has exhibited a contrasting trend—increasing during the early 2000s and stabilizing only since around 2007
(Chen et al., 2014b; Chen et al., 2021). $O_3$ formation is governed by highly nonlinear chemical reactions involving its



precursors, nitrogen oxides ($NO_x$) and volatile organic compounds (VOCs). The estimation of $NO_x$ emissions is relatively well-characterized, due to their simpler composition and reliable monitoring from over 80 air quality monitoring stations across Taiwan, VOC emissions present a far greater challenge.

VOCs comprise hundreds of chemically diverse species, each with distinct reactivities and emission sources. This complexity makes accurate VOC emission estimation and model validation particularly difficult. Current regional air quality models (AQMs), including chemical transport models (CTMs), reduce computational demand by grouping chemically similar VOCs into lumped surrogate species. As a result, many individual VOCs and their unique roles in $O_3$ formation remain unresolved in model simulations. Furthermore, most of Taiwan's MOENV monitoring sites report only total mixing ratios, lacking speciated VOC data necessary to validate individual model species.

Our current MOENV monitoring network monitors VOCs at approximately half of the 80 sites, but only provides total mixing ratios of non-methane hydrocarbons (NMHCs), making it difficult to validate the mixing ratios of individual VOC simulations and their subsequent impact on $O_3$ formation.

Following the 1990 amendments to the Clean Air Act in the United States, Photochemical Assessment Monitoring Stations (PAMS) were established in ozone non-containment areas. These stations provide detailed observations of individual VOC

species. Taiwan adopted a similar version of PAMS in 2001, using multiple automated gas chromatographs (auto-GC) as the core observation tool (Wang et al., 2004). PAMS stations in Taiwan measure up to 54 individual VOC species, providing critical hourly data for understanding the sources of pollution and offering direct evidence of the photochemical processes contributing to $O_3$ formation.

Since the availability of PAMS, many researchers have used these data to investigate the relationship of individual VOC

species with $O_3$. In 1995, Cardelino and Chameides (Cardelino and Chameides, 1990) analyzed the Relative Incremental Reactivity (RIR) of 54 VOC species measured by PAMS, classifying these VOCs and assessing the contributions of anthropogenic and biogenic sources to $O_3$ formation. In Taiwan, Yang et al. (Yang et al., 2005) analyzed continuous PAMS data from central Taiwan, revealing diurnal and seasonal variability in VOC species, which were further classified into petroleum leaks, vehicular emissions, and biogenic sources. Lee and Wang (Lee and Wang, 2006) also studied isoprene,

identifying its seasonal variability and its relationship to diurnal $O_3$ fluctuations. Ahmed et al. (Ahmed et al., 2025) reported trends of PAMS species over two decades of measurements in Texas, highlighting an increase in isoprene attributed mainly to rising temperatures over time. However, most related studies only focused on individual VOC measurements without integrating the overall PAMS data with VOC simulations. Theoretically, ozone simulations require the incorporation of numerous key reactive species and their associated chemical reactions. However, due to computational limitations, current

state-of-the-art air quality models (or chemical transport models) such as CMAQ and GEOS-Chem (https://gmao.gsfc.nasa.gov/) employ a simplified chemical mechanism, where chemically similar species are aggregated into surrogates to reduce computational load (Zhang et al., 2005). Commonly used lumped mechanisms include CBM (Zaveri and Peters, 1999), SAPRC (Carter, 2010), RADM2 (Stockwell et al., 1990), and RACM (Stockwell et al., 1997). While these mechanisms significantly reduce simulation resource consumption, they obscure the representation of individual



species and their specific reactions and limit the ability to compare modeled results with detailed speciated VOC observations, making it challenging to directly simulate individual VOC species with trustworthy results. As previous studies have shown, while PAMS stations have provided continuous VOC data for years, air quality models have yet to fully simulate the detailed variability of individual VOC species. Only in recent years have models begun to assess discrepancies between simulated VOC emissions and observations, leading to efforts to refine and update VOC emission inventories and

source profiles (Chen et al., 2010; Ying and Li, 2011; Chen et al., 2014a; Chen et al., 2015; Su et al., 2016). (Ge et al., 2024; Rowlinson et al., 2024).

This study aims to simulate pollution episodes using the CMAQ model in conjunction with PAMS observations to evaluate the model's capability in accurately representing VOC species. Specifically, the focus is on assessing how VOC simulations align with PAMS observations by employing a modified version of CMAQ, referred to as CMAQ-PAMS, which is designed

to simulate individual PAMS species. By comparing the model results with PAMS observations, the study seeks to enhance VOC species simulations and refine emission inventories, ultimately quantifying the aggregated PAMS contribution to ozone formation

## 2 CMAQ-PAMS model development

### 2.1 Chemical Mechanism Implementation

**Fig. 1** shows the CMAQ modeling framework revised for CMAQ-PAMS (red parts). The chemical mechanism initialized for the 54 PAMS species is the Carbon Bond (CB) chemical mechanism (in the version of CB05e51 and CB6r3, under CMAQ v5.2 model structure; two versions of CB are revised for PAMS species, so-called CB05e51pams and CB6r3pams). The 54 PAMS species are found in CB05e51, i.e., ETHA (ethane), ETH (ethene), and ISOP (isoprene), and CB6r3, i.e., ETHA, ETH, ISOP, PRPA (propane), ETHY (ethyne), respectively.

The chemical reactions for the 54 PAMS species are assigned from the related lumped species (PAR, OLE, OLI, TOL, XYL, ETH, ETHA, ISOP, PRPA, ETHY, and BENZ, respectively). The PAMS species are primarily primary pollutants, and their chemical losses mainly occur through reactions with hydroxyl (Knote et al.) , $O_3$, and nitrate ($NO_3$) radicals. The reaction rates for PAMS species can be found in the work of Chen et al. (Chen et al., 2010), where the PAMS chemical mechanism was first developed.

The PAMS chemical mechanism is connected to the CB chemical mechanism in parallel; however, it does not contribute to ozone simulations. Instead, the PAMS species are assigned their own individual reaction constants for reactions with OH(Knote et al.), $NO_3$, and $O_3$ to account for loss. This allows for comparisons with observed PAMS data.





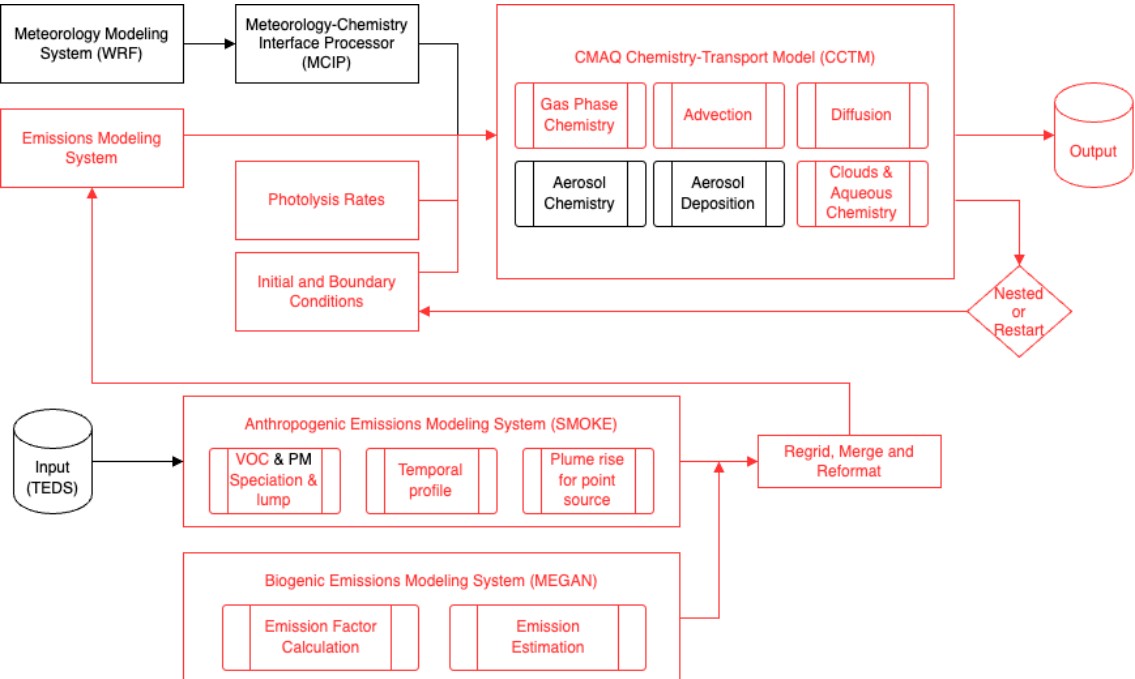

**Figure 1: Model implementation for CMAQ-PAMS (in red).**

## 2.2 CCTM revision

The UTIL modules in the CMAQ model are used to generate new chemical mechanisms, specifically for creating the CCTM module to simulate PAMS species. In the chemical mechanism compiler (chemmech), the PAMS species are put into the species name list (GC_cb6r3pams_ae6_aq.nml), and the additional chemical reactions are added with a new list of chemical reactions for PAMS species (mech_cb6r3pams_ae6_aq.def). These files are inputs for creating a new CCTM module to include PAMS species.

The Euler Backward Interactive (EBI) solver (create_ebi) is used for solving the new CB chemical mechanism with PAMS species. Also, the in-line photolysis option is chosen for calculating photolysis rates (inline_phot_preproc) to provide a set of molecular absorption CSQY data files consistent with the photolysis reactions in the new CB chemical mechanism with PAMS species. Finally, the new aqueous chemistry is merged with a pre-existing cloud module (acm_ae6) with updated Henry constants for the PAMS species (acm_ae6_pams). Altogether, the in-line photolysis rates and the new chemical mechanism with PAMS species are put into the MECHS module, the EBI solver is put into the gas module, and a revised aqueous chemistry is put into the cloud module.



## 2.3 Emission preprocessor for PAMS species

The emission preprocessor system for the CMAQ-PAMS modeling system (**Fig. 1**) followed the procedures of Sparse Matrix Operator Kernel Emissions (SMOKE) to process gas-phase and aerosol emissions. For CMAQ-PAMS modeling, there are mainly two code changes needed for adding PAMS species emissions: one is to add the 54 PAMS species array into the current CB emission array, and the other is to add the PAMS species speciation code into the current CB speciation profile for lumped VOC species. The original emission arrays for CB05e51 and CB6r3 included 46 and 48 species (including aerosols), respectively; now, after adding the 54 PAMS species, they include up to 100 and 102 species.

The VOC speciation data (GSPRO/GSCNV) are generated to provide VOC speciation profiles for the user-defined chemical mechanism for executing the emission preprocessor system (SMOKE). Before the VOC individual organics are lumped into lumping groups, the SPECIATE Tool (now moved to S2S-Tool) provides the linkages between separate organic gas and their speciation rates for each emission source (similar for aerosols and mercury). After the VOC speciation data is set, the emission preprocessor system follows the speciation profiles to divide the total VOC (NMHC) emission amounts into lumped groups. For CMAQ-PAMS, the VOC speciation data are updated in CB speciation profiles with PAMS source speciation details (gspro.cb05.pams and gspro.cb06.pams). While processing the emission inventory from the total to the gridded hourly amount, these additional PAMS speciation profiles (originally not in SMOKE therefore need to be summarized from the VOC speciation database) are used to subtract PAMS species emissions and put into the PAMS species emission array in the emission preprocessor system. After that, the PAMS species emissions abide by the consistent procedures of the emission preprocessor system to assign the temporal profiles, calculate plume rise for point sources, put geolocated emissions into the grid, and merge/reformat anthropogenic and biogenic emissions together. The biogenic emissions for the CMAQ-PAMS modeling system are generated by the Model of Emissions of Gases and Aerosols from Nature (MEGAN) modeling system (Guenther et al., 2006; Guenther et al., 2012). Considering isoprene as the only species whose biogenic emission dominates the total isoprene emissions (anthropogenic + biogenic emissions), as isoprene (ISOP) has already been separated for CB. As a result, no code revision is needed for the CMAQ-PAMS on biogenic emissions.

## 3 Model simulation setup

### 3.1 Model configuration

The WRF meteorological model version 3.8 (Skamarock et al., 2008) and CMAQ-PAMS model (based on CMAQ version 5.2 (Byun and Ching, 1999; Byun and Schere, 2006)) were used in this study to conduct meteorological and air quality modeling. The physical and chemical configurations of the WRF-CMAQ system are based on the real-time air quality forecasting (AQF) system in Taiwan (Cheng et al., 2021), with an offline approach, two-level nesting domains from the East Asia coarse domain (15 km, D1) to the nested domain over Taiwan with a 3-km spatial resolution (D2), with 30 non-uniform





sigma layers from the surface (a thickness of approximately 20 m) to the top of the atmosphere. Anthropogenic emissions are from the 2010 Model Inter-Comparison Study for Asia (MICS-Asia) emission inventory (Li et al., 2017) with the projection of China's anthropogenic emissions (Zheng et al., 2018) (D1) and the TEDS version 11.0 (D2). Biogenic emissions are estimated by MEGAN version 2.0.4 (Guenther et al., 2012).

Assuming the transboundary influence for the PAMS species is limited, the programs used for preparing chemical Initial CONditions (ICON) and Boundary CONditions (BCON), BCs for PAMS species are set as zero. In other words, the background concentrations of PAMS species from East Asia are neglectable, and the mixing ratios are mainly the result of domestic emissions on the island.

### 3.2 Case study background and model evaluations for critical pollutants

Since the development of the CMAQ-PAMS model, multiple case studies have been conducted to evaluate its performance under different synoptic conditions. Initial simulations were tested for events in November 2019 and September 2020 to verify the model's baseline stability and emission configurations under distinct meteorological scenarios. However, the primary analysis in this study focuses on the ozone pollution episode from September 27 to October 3, 2021 (**Fig. S1**), which provides a representative case with comprehensive data coverage and suitable photochemical conditions for model evaluation.

The 2021 episode was influenced by two tropical systems: Typhoon Mindulle (No. 16) and Tropical Storm Kompasu (No. 18). Although neither storm made direct landfall in Taiwan, their movement created a stable high-pressure system over the island. This synoptic pattern resulted in low wind speeds and suppressed atmospheric mixing, which promoted the accumulation of ozone precursors. Despite occurring in early autumn, a period typically less favorable for photochemical activity, the lowered mixing layer heights effectively enhanced ozone production through increased precursor concentrations. High ozone levels were observed in central and southern Taiwan throughout the period, while northern Taiwan experienced elevated $O_3$ between September 29 and October 10 (see **Fig. S2**).

Given the meteorological stagnation, regional representativeness, and elevated ozone levels, the 2021 case was selected for detailed model evaluation. The model's performance across all Taiwan Air Quality Monitoring Network (TAQMN) stations in western Taiwan is summarized in **Table S1**. Time series comparisons of wind fields and key pollutants ($NO_x$, VOCs, and $O_3$) at selected stations near PAMS sites are provided in **Fig. S3**. Statistical definitions and evaluation metrics are described in the Supporting Information.

### 3.3 Computing resources

The computing resources for executing CMAQ-PAMS were evaluated using a representative five-day run. The parallel computing environment was CentOS Linux version 7.4.1708 with Intel® Xeon® Silver 4110 CPU @ 2.10 GHz (8-Core), 28



nodes, and 64 GB RAM, with 56 nodes utilized for this case. The tested software environment included Intel compiler
19.0.1.144, ioapi 3.1, netCDF 4.1.3, and MPI 2.1.2.

Details of computing resources for the five-day scenario using CMAQ and CMAQ-PAMS are summarized in **Table 1**.
CMAQ-PAMS outputs 275 species, including 54 individual PAMS VOC species. For direct comparison, the default CB05
VOC species (e.g., ethane, ethylene, and isoprene, represented as ETHA, ETH, and ISOP) were retained under the same
nomenclature in both mechanisms. The CMAQ-PAMS simulation required approximately 1.8 times the CPU user time and
1.7 times the wall time compared with the standard CMAQ run. The additional 54 PAMS species increased the CONC file
size by ~3 GB relative to CMAQ.

**Table 2: Computing resource comparison between CMAQ and CMAQ-PAMS for a five-day scenario.**

|  | **CMAQ** (cb6r3_ae6) | **CMAQ-PAMS** (cb6r3pams_ae6) |
|---|---|---|
| Number of CONC species | 219 | 275 |
| CPU user time (h) | 58.07 | 103.77 |
| Wall time (h) | 2.16 | 3.72 |
| Data storage per CONC file (GB) | 12 | 15 |

## 4 PAMS network in Taiwan

The PAMS operation is based on a model first introduced in the United States in 1990 as required by the Clean Air Act
(https://www.epa.gov/laws-regulations/summary-clean-air-act). These stations employed auto-GC systems as the core
equipment to perform hourly measurements of 56 individual compounds with carbon numbers from $C_2$-$C_{12}$ in the U.S. The
compound list was shortened to 54 species by giving up the last two $C_{12}$ species of undecane and dodecane in Taiwan due to
the severe wall effect and residual problem in the analysis. For each hourly measurement, a 400 mL aliquot of the air sample
is drawn into the auto-GC for enrichment within an electronically cooled sorbent trap. The trap is then rapidly heated to
desorb the VOC analytes, which are back-flushed by a stream of high-purity helium used as the carrier gas. The analytes are
directed into two capillary columns with different stationary phases (PLOT and BP-1) to ensure optimal separation of the
species. Two flame ionization detectors (FID) are employed for the two columns (Wang et al., 2004).
In 2001, MOENV of Taiwan adopted the U.S. PAMS program, which has been in operation ever since. Over the years, the
program has expanded to include 15 stations located across the western part of the island, where more than 95% of the
population resides. This comprehensive dataset, characterized by a lengthy monitoring period and a significant number of
stations over a relatively small geographical domain, provides valuable insights into identifying pollution sources and offers
direct evidence of the photochemical processes that lead to the formation of $O_3$. Each PAMS station is strategically placed
next to a TAQMN station to provide hourly observations of meteorological conditions and criteria air pollutants.



There were 12 PAMS stations employed for the September 2021 case study based on data completeness (**Fig. 2**): Wanhua (W-site), Tucheng (T1-site), Pingzhen (P1-site), Zhongming (Z1-site), Zhushan (Z2-site), Taixi (T2-site), Puzih (P2-site), Tainan (T3-site), Qiaotou (Q-site), Xiangnag (X-site), Chaozhou (C-site), and Dacheng (D-site). Observations of individual VOC species from 12 PAMS stations across Taiwan will be directly compared with the CMAQ-PAMS model simulation
results.

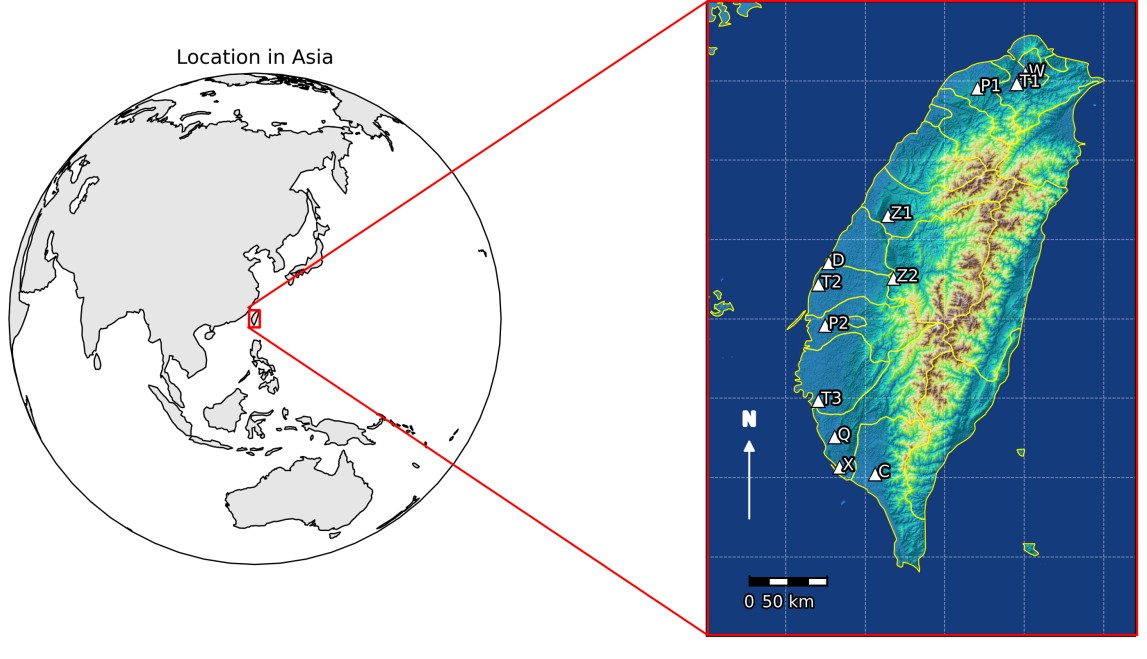

**Figure 2: Geolocations of 12 PAMS stations in Taiwan. Shaded areas indicate terrain heights (Indigo denotes the ocean. On land, elevation increases from green to yellow to brown with hill-shaded relief).**


## 5 Results and Discussion

### 5.1 Comparison of PAMS modeling results with observations

#### 5.1.1 Individual species

**Fig. 3** presents the time series of the observations vs. simulation results at the T-site in northern Taiwan from September 27
to October 3, 2021. The parameters evaluated include the mixing ratios of the summation and selected PAMS VOCs, measured in ppbC (normalized to methane in carbon number), such as TPAMS, n-pentane, ethylene, toluene, and 2-methylhexane (All other PAMS sites exhibited similar results for PAMS species simulations, as shown in **Fig. S4**).



When analyzing the PAMS species simulations, our first criterion is to evaluate both the simulated and observed wind fields to confirm that the atmospheric circulation patterns are true to the environment. Notably, model simulations clearly captured

the significant change in wind direction for any simulation period, as shown in **Fig. 3** for a case study from September 29 to October 1. The diurnal variations of the PAMS species illustrate the effectiveness of the physical and chemical processes estimated through CMAQ-PAMS modeling. However, the discrepancies between the simulations of PAMS species and actual observations highlight the inaccuracy in the emission estimates for these species, despite showing similar diurnal patterns (**Fig. 3**). Some species are either underestimated or overestimated, with some even reporting zero emissions. This

discrepancy was noted when the PAMS species modeling technique was first introduced (Chen et al., 2010). The primary cause of this inconsistency stems from the mismatch in using U.S. VOC speciation profiles to represent the local organic sources on the island.

While improving emission inventories and speciation profiles could theoretically help close the gap, doing so requires tremendous effort with uncertain outcomes. Instead, we adopted a reverse modeling, or top-down, approach [*Chen et al.,*

*2014c*] to calibrate PAMS species emissions using observations from 12 PAMS monitoring sites as the "ground truth". After adjustment, the simulated mixing ratios (ModSIM) became more consistent with the observations, as indicated by stronger correlations between simulated and observed PAMS species. For example, during the case study periods in the autumn of 2021, the median predicted-to-observed (P/O) ratio improved from 0.66 in OrigSIM to 1.05 in ModSIM. **Fig. 3** exemplifies four species for four types of simulation outcomes: comparable, overestimated, underestimated, and zero adjustment. For *n-*

*pentane*, the simulated and observed results are in agreement; thus, no adjustments are needed. In the case of *ethylene*, OrigSIM (shown in red) significantly overestimates its mixing ratios, with numerous peaks not reflected in the observations. ModSIM (shown in green) effectively reduces these overestimations, leading to a much-improved agreement with observations throughout the period. In the case of *toluene*, OrigSIM substantially underestimates its concentrations, particularly in the middle portion of the time series. ModSIM demonstrates a marked improvement, closely matching

observed values. In the case of *2-methylhexane*, OrigSIM produces zero concentrations throughout the entire period due to the absence of emissions. ModSIM, on the other hand, significantly enhances the simulation and reasonably captures the observed mixing ratios. Some species, such as *cyclohexane*, *1-pentene*, and *styrene*, have emissions that are excessively high and require significant adjustments to align with observed levels, as shown in **Table 2**. Among the 54 PAMS species, 11 showed simulated concentrations comparable to observations, eight overestimated, 31 underestimated, and four species — *2-*

*methylhexane*, *p-ethyltoluene*, *o-ethyltoluene*, and *p-diethylbenzene* — displayed zero simulated values. Similar patterns were observed in other case studies conducted in 2019 and 2020 (results not presented for brevity). Overall, ModSIM represents a substantial improvement over OrigSIM in simulating PAMS species in ambient air. As a result, given the current limitations in updating the VOC speciation profiles to align with domestic emission conditions, utilizing PAMS hourly observations across the island offers an alternative to enhance emission estimations significantly. By employing the

top-down approach, these observations can effectively refine and calibrate the PAMS emission amounts, offering a more realistic representation of VOC emissions despite the lack of suitable updates to the speciation profiles.  Schneidemesser et



al. (Von Schneidemesser et al., 2023) demonstrated significant discrepancies between emission inventories and observed non-methane VOC levels, particularly due to outdated speciation profiles and missing oxygenated VOCs (OVOCs). The current study further supports these findings by demonstrating that traditional Source Classification Code (SCC)-based

speciation does not adequately capture urban VOC profiles. By incorporating PAMS-based speciation, CMAQ-PAMS significantly improves model performance and aligns better with real-world abundance.

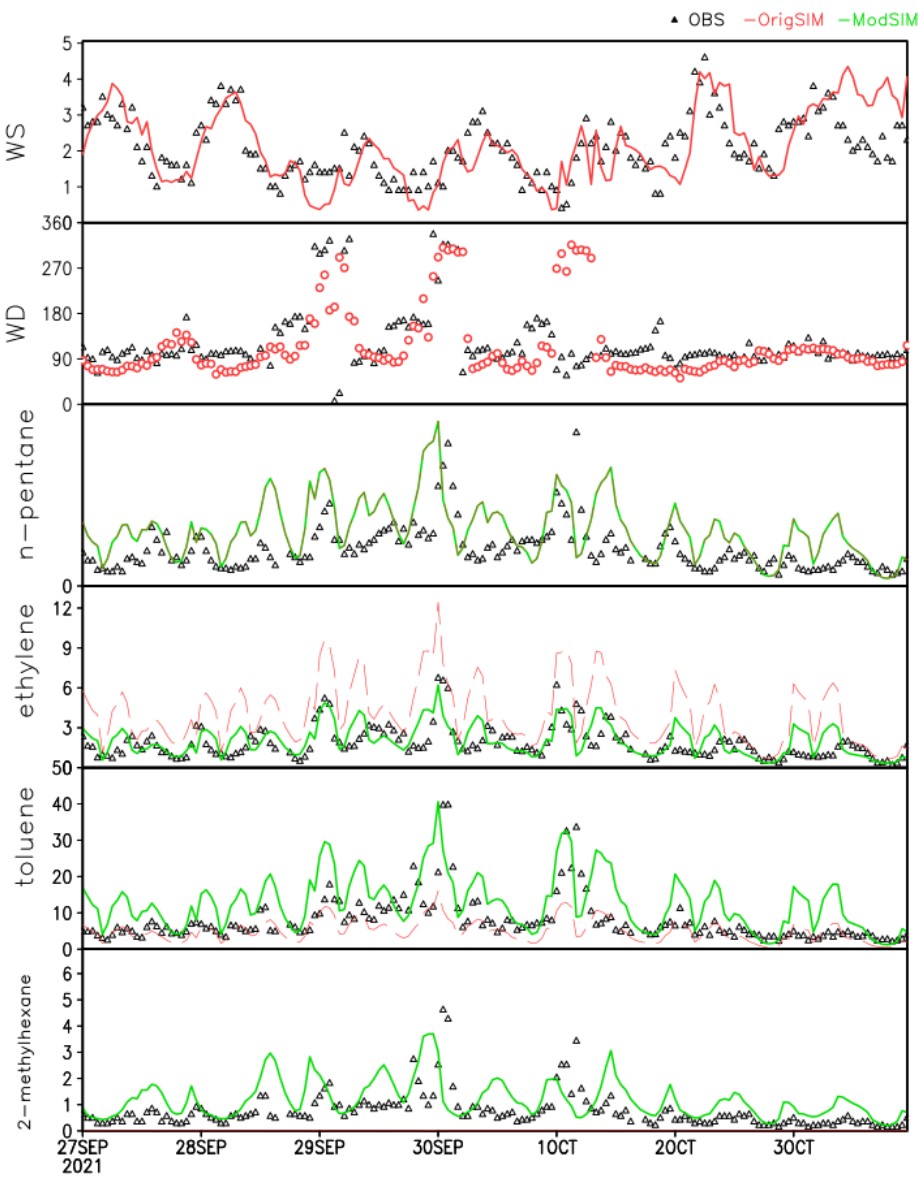

**Figure 3: Time series of wind fields and selected PAMS species simulations with observations at W-site (northern Taiwan). The**
**black triangles are PAMS observations, and the red and green solid lines are PAMS simulations before and after the PAMS species emission adjustment—time labels at 0:00 CST. Units are ppbC.**





**Table 2: PAMS species emission adjustment factors for CMAQ-PAMS.**

| Group | Species | Adjustment Factor | Group | Species | Adjustment Factor |
|---|---|---|---|---|---|
| Alkanes | propane | 1.6 | Alkenes | trans-2-butene | 2.91 |
| | isobutane | 3.28 | | cis-2-butene | 1.91 |
| | n-butane | 0.75 | | trans-2-pentene | 1.64 |
| | 2,2-dimethylbutane | 2.55 | | cis-2-pentene | -- |
| | isopentane | 1.68 | | propylene | -- |
| | n-pentane | -- | | 1-butene | -- |
| | 2,3-dimethylbutane | 1.45 | | 1-pentene | 0.41 |
| | 2-methylpentane | 1.57 | | isoprene | -- |
| | 3-methylpentane | 2.17 | | ethylene | 0.52 |
| | n-hexane | 1.38 | Alkynes | acetylene | 2.27 |
| | 2,2,4-trimethylpentane | -- | Aromatics | benzene | -- |
| | 2,4-dimethylpentane | 1.48 | | toluene | 2.54 |
| | 2-methylhexane | 2.48 | | ethylbenzene | 1.78 |
| | 2,3-dimethylpentane | -- | | styrene | 0.22 |
| | 3-methylhexane | -- | | isopropylbenzene | 0.96 |
| | n-heptane | -- | | n-propylbenzene | 1.9 |
| | 2,3,4-trimethylpentane | 0.51 | | m,p-xylenes | 3.47 |
| | 2-methylheptane | 3.05 | | o-xylene | 1.91 |
| | 3-methylheptane | 2.37 | | m-ethyltoluene | 2.67 |
| | n-octane | 1.56 | | p-ethyltoluene | 1.84 |
| | n-nonane | 2.86 | | o-ethyltoluene | 1.73 |
| | n-decane | -- | | 1,2,4-trimethylbenzene | 2.06 |
| | n-undecane | -- | | 1,2,3-trimethylbenzene | 3.34 |
| | ethane | 3.06 | | 1,3,5-trimethylbenzene | 0.56 |
| Cycloalkanes | cyclopentane | 4.13 | | m-diethylbenzene | 0.6 |
| | methylcyclopentane | 2.23 | | p-diethylbenzene | 1.49 |
| | cyclohexane | 0.38 | | | |
| | methycyclohexane | 1.73 | | | |

### 5.1.2 Summation of PAMS species (i.e., TPAMS)

Previously, the absence of PAMS observations and CMAQ-PAMS simulations made it difficult to simulate and validate individual VOC simulations directly. As mentioned earlier, the lumping technique prevents modeling from being directly compared with speciated VOC observations. Only a few isolated species, such as *ethane*, *ethylene*, and *isoprene*, could be directly matched with PAMS data. Moreover, even the total lumped VOC concentrations did not align well with the observed total VOC concentrations (Chen et al., 2015). The development of the CMAQ-PAMS model now allows for direct validation of both speciated VOC simulations and total PAMS (TPAMS) concentrations by summing up the 54 PAMS species. It is worth noting that the TAQMN network also monitors total ambient VOCs using a non-methane hydrocarbon (NMHC) analyzer, which can be used to validate total VOC (TVOC) simulations. However, this instrument, equipped with a flame ionization detector, is more sensitive to straight-chain hydrocarbons than to oxygenated hydrocarbons. As a result, it may theoretically underestimate total VOC concentrations to some extent. In **Fig. 3a**, the TPAMS is expected to be slightly





lower than the TVOC values observed by TAQMN, and certainly lower than the true ambient TVOC, since only 54 major NMHCs are detected, rather than the full spectrum of ambient VOCs (Chen et al., 2014b). **Fig. 3** presents a time series comparison of observed and simulated TPAMS at W-site from September 27 to October 9, 2021. The observed values (OBS) are compared with two simulation sets: the original simulation (OrigSIM, shown in red) and the adjusted simulation

(ModSIM, shown in green). Overall, the modeled TPAMS diurnal patterns align well with observed variations, though the model fails to capture some extreme values. This discrepancy arises because PAMS species emissions in the emission preprocessor follow predefined temporal cycles (e.g., monthly, weekly, and daily), whereas real-world emissions can exhibit irregular fluctuations. The model effectively simulates the evolution of organic species under typical atmospheric conditions, but is unsuitable for accounting for unusual emission events. Despite these limitations, ModSIM overall matches the

observed averages and demonstrates a significant improvement over OrigSIM. This improved performance is consistent across all **12 monitoring stations**, successfully correcting the over- and underestimations observed in OrigSIM (**Fig. S4**). The limited improvement in correlation (R) between the original and modified simulations and the observations can be attributed to the type of adjustments applied in the modified simulation. When the PAMS emissions were updated over time, no changes were made to the temporal cycles of the individual species. These cycles are crucial for determining the temporal

variability of simulated concentrations and hence their correlation with observations. Therefore, even though the emission magnitudes were correct, the lack of proper adjustment in the emission temporal characteristics for individual species limits the model's ability to capture the observed temporal patterns. This led to only marginal improvements in the correlation (R) between the simulations and the observations.

While the top-down approach of PAMS species emission modification can successfully improve the model simulation

results to represent the evolution of the observed values better, one still needs to roll back to consider the possible causes of this misrepresentation. Since the VOC speciation processes are based on U.S. VOC speciation profiling, the mismatch of the VOC profiles for the sources will lead to biased speciated VOC emission amounts. As an example, the **Supporting Information (Fig. S5)** illustrates uncertainties arising from erroneous VOC speciation profiling, using household emissions in Taiwan as a case study.

Overall, the island-wide PAMS dataset, featuring hourly resolution and extensive spatial coverage, serves as a reliable ground truth, enhancing the robustness of model evaluations in both spatial and temporal representativeness.

## 5.2 Dominant PAMS species and their distributions across the island

While both $NO_x$ and VOCs are precursors to the $O_3$ formation, $NO_x$ consists only of NO and $NO_2$ and thus is not

discriminative between sources. Conversely, VOCs include numerous organics and are different in composition between sources; therefore, each source type may have a composition profile unique to itself. Due to source characteristics and photochemical reactivities of species, the relative abundance among species changes with time and space.




After the PAMS emission has been modified, **Table 3** provides a comparative list of the top 20 individual VOC emissions
from anthropogenic sources in Taiwan, including point, line, and area sources. The VOCs are ranked according to their
emission quantities from each source type, highlighting the significant contributors to overall VOC emissions. The top 10
VOC species, such as *toluene*, *xylenes*, *propylene*, *ethylene*, *formaldehyde*, *1,3-butadiene*, etc., dominate point sources to
contribute 58% to total VOC emissions.

The comparison of VOC emissions from point, line, and area sources reveals that *toluene* is the largest individual VOC
emitted across all three source types. When modifying the PAMS species emissions, adjustments were primarily based on a
linear relationship between the modeled values and observed data at each grid point, while the total VOC amount is
conserved.

**Table 3: List of top 20 VOC emissions from point, line, and area sources.**

| No. | Point species | emission | Line species | emission | Area species | emission |
|---|---|---|---|---|---|---|
| 1 | Toluene | 37,042 | Toluene | 48,827 | Toluene | 137,550 |
| 2 | Propylene | 14,359 | P-xylene | 32,516 | 1,3-butadiene | 92,359 |
| 3 | M-xylene | 12,942 | 1,2,4-trimethylbenzene | 31,730 | Propylene | 89,888 |
| 4 | Ethylene | 12,814 | 3-Ethyltoluene | 23,394 | Ethylene | 80,292 |
| 5 | Formaldehyde | 10,683 | Trans-2-butene | 23,253 | Isomers of xylene | 69,426 |
| 6 | Isomers of xylene | 8,882 | 1,2,3-trimethylbenzene | 22,286 | Formaldehyde | 61,055 |
| 7 | O-xylene | 7,910 | O-xylene | 20,843 | M-xylene | 34,378 |
| 8 | 1,3-butadiene | 7,159 | 1,3-butadiene | 19,796 | 1-butene | 30,697 |
| 9 | Methyl methacrylate | 4,126 | Ethylene | 19,255 | O-xylene | 14,034 |
| 10 | Acrolein (2-propenal) | 3,472 | Isopentane | 19,061 | Acetaldehyde | 13,027 |
| 11 | Acetaldehyde | 3,424 | Formaldehyde | 12,675 | Methyl methacrylate | 11,545 |
| 12 | P-xylene | 3,406 | Cis-2-butene | 12,673 | Propane | 11,409 |
| 13 | Acrylic acid | 2,963 | Propylene | 12,575 | Ethyl acrylate | 10,906 |
| 14 | 1-butene | 2,793 | 2-methyl-2-butene | 12,344 | N-butane | 10,894 |
| 15 | Methyl acrylate | 2,621 | Acetaldehyde | 11,595 | Benzene | 10,163 |
| 16 | Ethyl alcohol | 2,544 | Trans-2-pentene | 11,133 | Acrolein | 9,651 |
| 17 | Ethyl acrylate | 2,481 | 1,3,5-trimethylbenzene | 10,883 | Ethylbenzene | 9,551 |
| 18 | Methyl isobutyl ketone | 2,460 | N-butane | 10,355 | P-xylene | 9,357 |
| 19 | Isoprene | 2,410 | 1-butene | 8,594 | Propylene glycol | 9,108 |
| 20 | Ethylbenzene | 2,368 | Isobutane | 7,894 | Acrylic acid | 8,269 |

The units of emissions are Mt/yr.


**Fig. 4** presents the average concentrations of VOC groups from PAMS simulations and observations at monitoring stations
for the selected 2021 case. The ModSIM results demonstrate a marked improvement over OrigSIM, particularly for the
alkane and aromatic groups (species-level comparisons are provided in **Table S2**). For TPAMS, the average concentration
increased from 39.61 ppbC in OrigSIM to 68.39 ppbC in ModSIM, bringing it much closer to the observed 63.94 ppbC. The
original simulation tends to underestimate alkanes, whereas the modified simulation better captures the total magnitude in
alignment with observations. Notably, acetylene—underrepresented in OrigSIM but not absent—has been corrected through
the top-down calibration approach. Unlike the alkane, alkene, and aromatic groups, which include multiple PAMS species,





the alkyne group in **Fig. 4** consists solely of acetylene. This emission in the original emission inventory is under-representative, as acetylene serves as a key tracer for incomplete combustion (see **Section 5.4**). The adjustment in ModSIM

helps recover this loss and brings simulated values into closer agreement with observations, improving overall mass conservation.

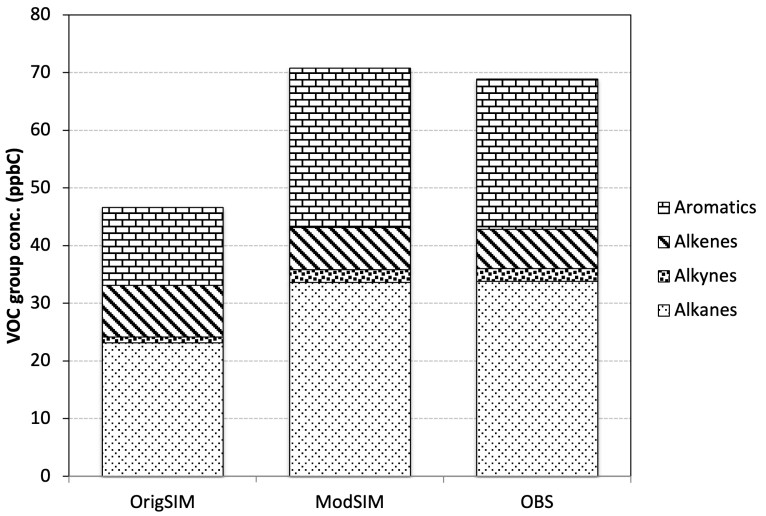

**Figure 4: Averages of PAMS observations and simulations by functional groups.**


## 5.3 In-depth diagnosis of ModSIM by CMAQ-PAMS

Most PAMS species have shown notable improvements following emission adjustments. A more in-depth demonstration of the enhanced model performance can be illustrated through two representative species: *toluene*, a typical anthropogenic VOC primarily emitted from solvent use and vehicular sources, and *isoprene*, a dominant biogenic VOC released from

vegetation. These species not only represent contrasting emission origins but also play distinct roles in ozone formation chemistry. When applying modeling to real-world scenarios, ensuring realistic and accurate temporal and spatial distributions is critical for effectively identifying underlying issues and informing appropriate mitigation strategies.

*Toluene*, being the most abundant compound observed across the PAMS network and exhibiting the highest emissions from point, line, and area sources (**Table 3**), warrants a detailed evaluation of model performance throughout the island. For the

point source, *toluene* is the most used solvent and an important feedstock in the industry. For line sources, four-stroke motorcycles and light-duty gasoline passenger vehicles are the primary contributors to *toluene* emissions. For area sources, the main contributor is surface painting or coating of industrial products category, which includes emissions related to solvents used in commercial and residential activities. **Fig. 5a** presents a comparison of observed and simulated *toluene* concentrations over time at all PAMS stations. The original simulations consistently underestimated *toluene* levels, with



predicted-to-observed (p/o) ratios around 0.5 (results not presented). Emission adjustment would require an increase by a factor of 2.54 to reach the observed island-wide averages (refer to **Table 2**). The time series of *toluene* mixing ratios across all PAMS sites indicates that the model generally captures the overall observed levels and distinct diurnal cycles. However, discrepancies remain in some peak values, which may result from inaccuracies in the emission inventory or limitations in some local sources as described earlier. The temporal alignment of concentration peaks suggests that the model reasonably captures diurnal variation patterns.

(a) toluene

(b) isoprene








**Figure 5: Time series of (a) toluene and (b) isoprene at PAMS sites across the island (ppbC). Black triangles are PAMS observations, and green/red solid lines are CMAQ-PAMS simulations (with/without PAMS emission modification)—time labeled at 0:00 CST.**

**Fig. 6a–b** presents the spatial mapping of *toluene* emissions and simulated concentrations at 10:00 CST on September 28, 2021, alongside the corresponding wind field. The emission mapping reveals that high *toluene* emissions are concentrated in urban and industrial areas, particularly in the western part of the island, where population density, traffic, and industrial activity are the most prevalent. These correspond to major area, line, and point sources. The wind field on that day indicates that local circulation was the primary driver of pollutant transport, leading to elevated simulated concentrations downwind of

major emission sources. However, noticeable mismatches in some regions between high-emission zones and corresponding simulated concentrations suggest that wind patterns significantly reshaped the spatial distribution of *toluene* concentrations. *Isoprene*, the only PAMS species primarily of biogenic origin, exhibits distinctly unique diurnal and seasonal variations that differ markedly from those of the other 53 PAMS compounds. Due to its extremely high reactivity as an essential precursor of ozone and secondary organic aerosols (SOA), and ubiquitous presence across the island, where approximately 70% of the





land surface is covered by vegetation, *isoprene* warrants an in-depth understanding through improved modeling approaches. Both anthropogenic and biogenic VOC emissions contribute to *isoprene* emissions; nevertheless, biogenic origin is thought to contribute the majority of the ambient abundance. The *isoprene* from anthropogenic emissions follows the protocol of emission procedures (VOC speciation, temporal/spatial allocation), whereas that from biogenic emissions is derived from the MEGAN emission model (**Fig. 1**). The results highlight the model's ability to capture the diurnal variation of isoprene

concentrations (**Fig. 5b**), particularly the daytime peaks driven by biogenic emissions. However, some discrepancies between simulated and observed data —especially underestimations during peak emission hours (**Fig. 5b**)—suggest that the model may not fully capture the complexity of biogenic emissions, which are highly sensitive to environmental factors including vegetation type, land use, soil conditions, temperature, and solar radiation. Overall, the model performs reasonably well in simulating *isoprene* levels, but further refinement could be achieved by incorporating more detailed land-use and

vegetation data. For example, in **Fig. 5b**, the northernmost sites (W, T1, and P1) show lower observed *isoprene* concentrations than predicted by the model. This overestimation is likely due to outdated land-use and vegetation data, as rapid urbanization in these areas may have significantly reduced vegetation cover (with lower *isoprene* biogenic emissions). Updating biogenic emission inventories and land-use status using satellite-based retrievals may provide a more accurate representation of current environmental conditions, thereby improving the fidelity of *isoprene* modeling *[Palmer et al.,*

*2003]*. When simulating *isoprene* emissions, no adjustments were made to the biogenic emission factors in the emission modification. Despite its strong emissions during the warm seasons, *isoprene* remains at consistently low yet detectable levels on the western side of the island. This is attributed to its high photochemical reactivity and the limited vegetation cover, as much of the land is allocated for dense human settlement. These levels generally align with the PAMS measurements (**Fig. 6c–d**). In sharp contrast, the eastern side of the island exhibits significantly higher *isoprene*

concentrations, primarily driven by the abundance of vegetation and minimal urban development.

(a) toluene-emission                                    (b) toluene-simulation



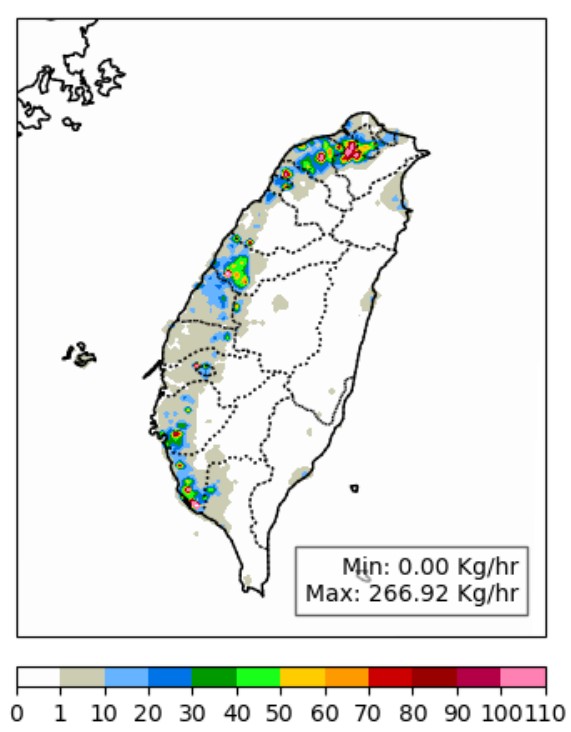

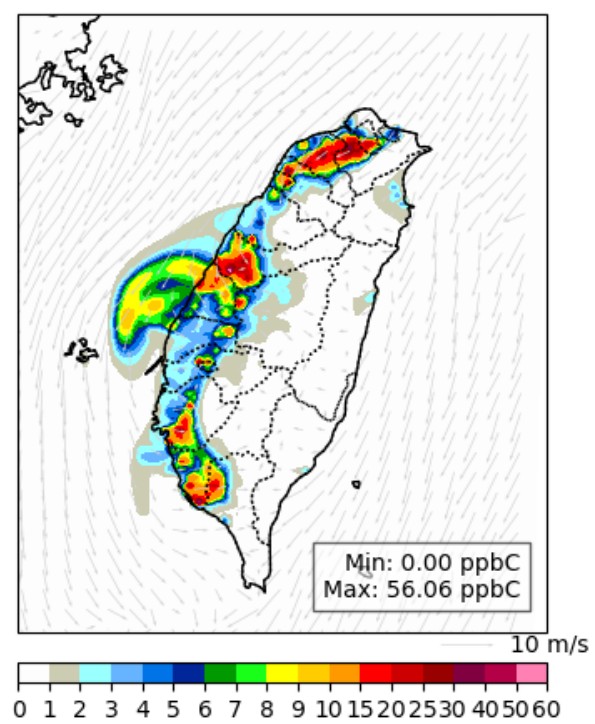

(c) isoprene-emission

(d) isoprene-simulation

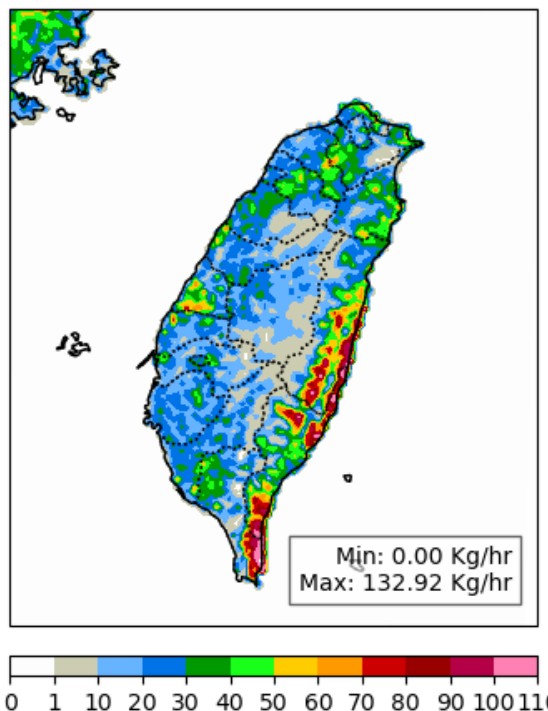

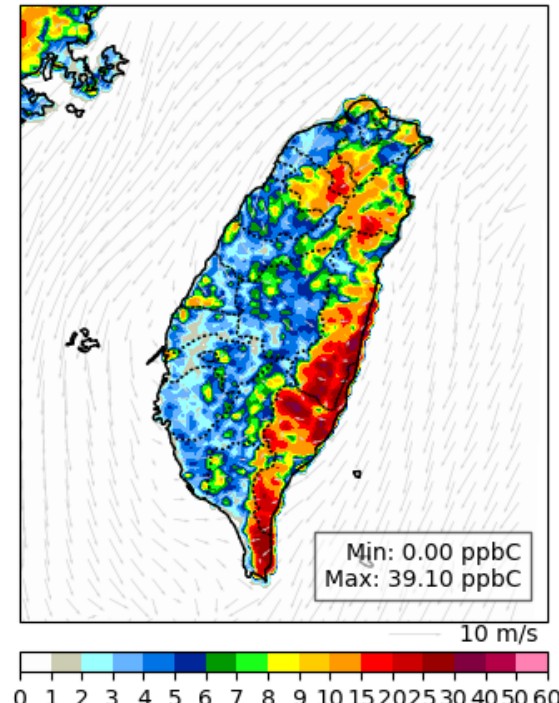





**Figure 6: Spatial mapping of toluene (a) emission (kg/hr) and (b) simulation (ppbC), and isoprene (c) and (d) along with the wind field across the island at 10:00 CST, 9/28/2021.**

(a)

(b)

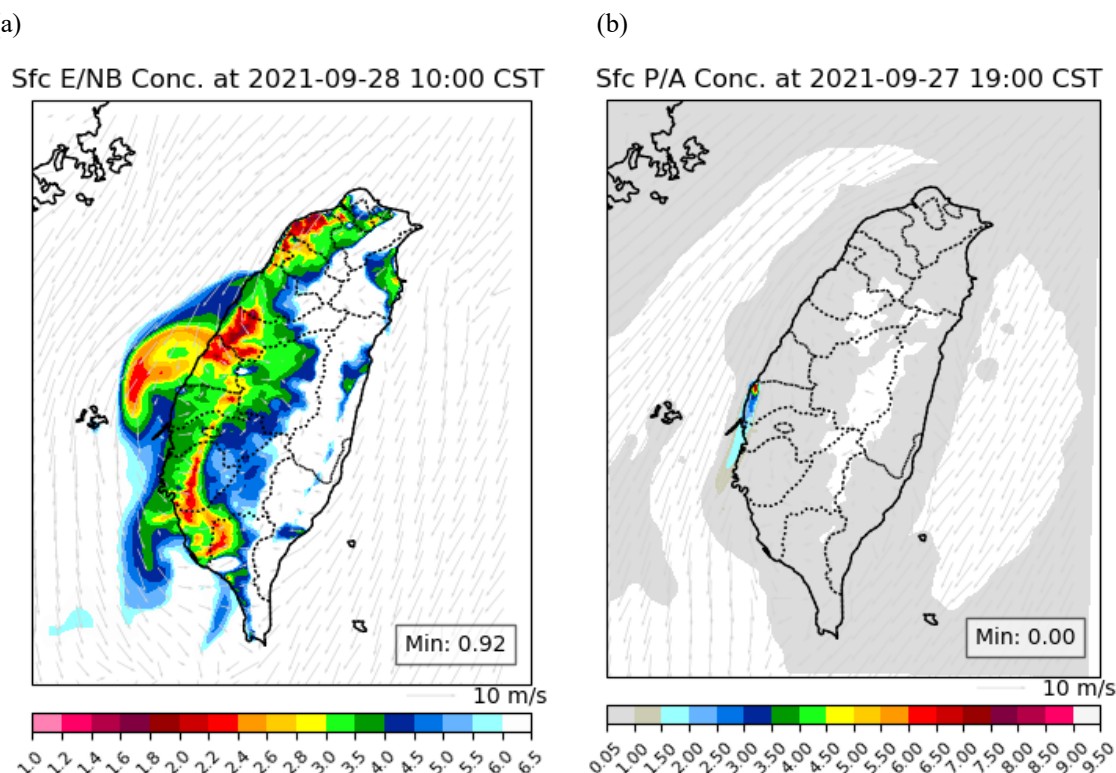


**Figure 7: Spatial mapping of modeled wind fields and indicative ratios of (a) ethane/n-butane (E/nB) at 10:00 CST, 9/28/3021, and (b) propylene/acetylene (P/A) at 19:00 CST, 9/27/2021.**

## 5.5 Accountability of PAMS emissions to ozone formation

In VOC-limited regimes, reducing VOC emissions is key to controlling $O_3$ formation. Understanding how PAMS species contribute to $O_3$ production is essential both for identifying mitigation targets and for justifying the value of the islandwide PAMS network. If PAMS species are found to significantly influence $O_3$ levels, targeting their sources could be an effective control strategy. The earlier calibration of PAMS species emissions against observed data provides a foundation for evaluating their area-specific contributions to $O_3$ formation across Taiwan.

**Fig. 8** presents the time series of observed and modeled VOC and $O_3$ concentrations during a 2021 episode, categorized by site types: W-site (urban), T1-site (urban/industrial), and P-site (urban/industrial). Model results are shown for total (including transboundary and background), domestic-only (TW), and PAMS-only emissions (TW_PAMS). Observed and modeled speciated PAMS values were drawn from nearby PAMS sites (locations in **Fig. 2**). Previous studies [Chen et al.,




2014b] showed strong coherence between observed total PAMS (TPAMS) and TVOC values, with TPAMS consistently
~30% lower due to limited speciation (54 NMHC species), while TVOC includes additional compounds such as OVOCs (Chen et al., 2015). To assess PAMS contributions to total VOCs and ozone, we examined TPAMS/TVOC ratios from three perspectives: emissions (EMIS), observations (OBS), and simulations (Ge et al.), as summarized in **Table 4** and **Fig. 8**. Emission data showed TPAMS accounted for 17–59% of TVOC, averaging 32%, indicating a substantial presence of non-PAMS VOCs (e.g., OVOCs and other unmeasured low-abundant species). In contrast, observations revealed higher PAMS
contributions, especially at X and T3 sites, where PAMS/TVOC approached 1.0, suggesting dominance of PAMS species in measured profiles. Intermediate ratios (0.51–0.64) at other sites were consistent with historical PAMS data (~0.7 average from 2007–2011) (Chen et al., 2014b), confirming the representativeness of current observations. However, simulated PAMS/TVOC ratios averaged 0.44, with higher values (≥0.50) at urban sites (W, Z1, T3, X), indicating improved model performance in populated areas. A notably lower ratio at T1 (0.34) may reflect model underrepresentation of OVOCs near
industrial sources.

O₃ levels over Taiwan result from both domestic photochemistry and transboundary influences, which vary seasonally and regionally (Chen et al., 2021; Choi et al., 2024; Lin et al., 2005). As **Fig. 9** shows, the modeled O₃ based on PAMS-only emissions (green lines) captures the timing of observed peaks but underestimates their magnitudes compared to full-VOC simulations (blue lines). PAMS only contributions accounted for 23–52% of total domestic O₃ production, underscoring their
importance in VOC-limited urban and industrial areas, yet also revealing the role of other VOCs, possibly OVOCs, as well as numerous low-abundant species not covered by the PAMS measurements. Overall, PAMS species represent a substantial fraction of total VOCs and play a significant role in shaping observed VOC profiles relevant to ozone formation. Bridging the gap between observed and modeled O₃ levels may require expanding the VOC speciation spectrum to include aldehydes and other major oxygenated VOCs (OVOCs). However, such enhancements would significantly increase the complexity and
cost of monitoring infrastructure, an issue beyond the scope of this study.

(a) W1-site (urban)

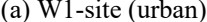

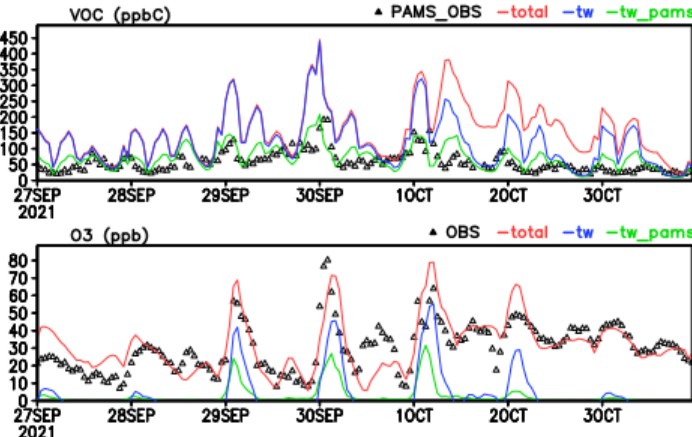





(b) T1-site (urban/industrial)

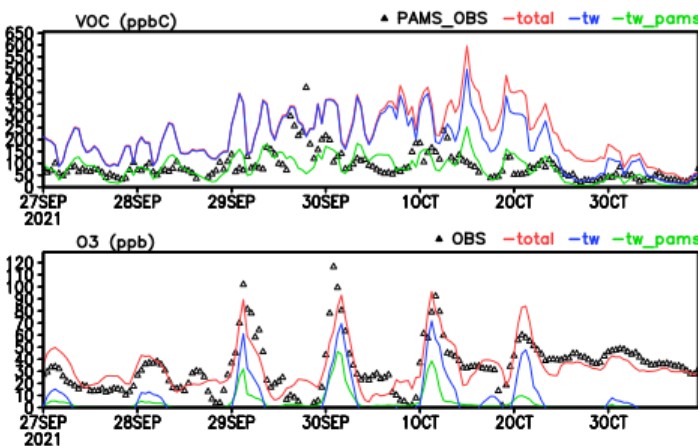


(c) P-site (urban/industrial)

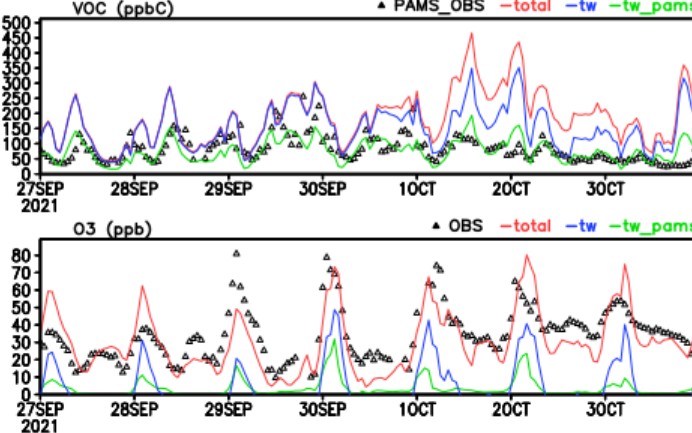

**Figure 8: Time series of observed and modeled VOCs and O₃ at selected TAQMN stations in the 2021 event from 9/27-10/03 (labeled at 8:00 CST). The black triangles are observations (from PAMS or TAQMN), red lines are simulations with total sources**
**(background, transboundary, and domestic emissions), blue lines are simulations with domestic emissions (PAMS and non-PAMS VOC emissions), and green lines are simulations with domestic PAMS VOC emissions only.**

**Table 4: Station-specific PAMS and total VOC amounts and their ratios across emissions, observations, and simulations, including associated modeled O₃ levels.**

| Station | VOC-EMIS | | | VOC-OBS | | | VOC-SIM | | | O₃-SIM | | |
|---|---|---|---|---|---|---|---|---|---|---|---|---|
| | PAMS | VOC | ratio | PAMS | NMHC | ratio | PAMS | VOC | ratio | PAMS | VOC | ratio |
| Wanhua (W) | 0.59 | 2.79 | 0.21 | -- | -- | -- | 78.61 | 155.69 | 0.50 | 6.33 | 13.78 | 0.46 |
| Tucheng (T1) | 0.42 | 2.47 | 0.17 | 86.83 | 136.56 | 0.64 | 90.45 | 225.73 | 0.40 | 12.75 | 24.35 | 0.52 |
| Pingzhen (P1) | 0.29 | 1.23 | 0.24 | -- | -- | -- | 86.45 | 186.46 | 0.46 | 8.32 | 18.31 | 0.45 |
| Zhongming (Z1) | 0.55 | 2.57 | 0.21 | 72.30 | 142.68 | 0.51 | 138.86 | 253.70 | 0.55 | 11.77 | 27.65 | 0.43 |
| Zhushan (Z2) | 0.10 | 0.24 | 0.41 | -- | -- | -- | 36.40 | 122.44 | 0.30 | 5.22 | 23.04 | 0.23 |
| Taixi (T2) | 0.07 | 0.12 | 0.59 | 57.93 | 91.80 | 0.63 | 34.79 | 101.37 | 0.34 | 6.28 | 12.22 | 0.51 |
| Puzi (P2) | 0.11 | 0.26 | 0.41 | -- | -- | -- | 46.65 | 117.80 | 0.40 | 5.71 | 18.27 | 0.31 |
| Tainan (T3) | 0.41 | 1.91 | 0.22 | 76.24 | 86.94 | 0.88 | 86.89 | 172.31 | 0.50 | 4.89 | 18.08 | 0.27 |



| | | | | | | | | | | | | |
|---|---|---|---|---|---|---|---|---|---|---|---|---|
| Qiaotou (Q) | 0.16 | 0.44 | 0.37 | -- | -- | -- | 54.78 | 119.03 | 0.46 | 4.77 | 18.33 | 0.26 |
| Xiaogang (X) | 0.37 | 2.13 | 0.18 | 97.05 | 97.31 | 1.00 | 87.91 | 162.16 | 0.54 | 5.10 | 15.54 | 0.33 |
| Chaozhou (C) | 0.10 | 0.21 | 0.46 | -- | -- | -- | 31.30 | 88.50 | 0.35 | 6.68 | 20.68 | 0.32 |
| Average | 0.29 | 1.31 | 0.32 | 78.07 | 111.06 | 0.73 | 70.28 | 155.02 | 0.44 | 7.07 | 19.11 | 0.37 |

**The units for VOC_EMIS, VOC_OBS, VOC_SIM, O$_3$_SIM are mole/s, ppbC, ppbC, ppb, respectively.
***The ratios of modeled O$_3$ are calculated for the morning O$_3$, considering local photochemistry.
--- Stands for sites not equipped with NMHC instruments.

## 6 Conclusions

This study presents CMAQ-PAMS, an enhanced air quality modeling framework that integrates speciated VOCs observed by Taiwan's PAMS network into the CMAQ system. By explicitly simulating 54 PAMS-targeted species, the model overcomes a major limitation of conventional lumped-species chemical mechanisms and allows direct comparison with observed VOC data.

Through a top-down calibration using islandwide PAMS observations, the modified model substantially improves agreement with observed VOC concentrations across chemical groups of alkanes, alkenes, and aromatics, and restored key tracers such as acetylene in the alkyne group. These refinements yielded more realistic spatial distributions and source contributions, improving the model's ability to represent VOC-limited O$_3$ formation regimes.

The calibrated PAMS species account for only ~32% of total VOC emissions yet contribute disproportionately (up to 52%) to domestic O$_3$ production, underscoring the importance of a few highly reactive species in driving ozone pollution. The integration of VOC indicator ratios further enhanced diagnostic capability for source identification and atmospheric processing.

CMAQ-PAMS serves as a robust, observation-constrained platform for improving chemical accuracy in regional ozone modeling. Future efforts should prioritize updating VOC speciation profiles with localized measurements, expanding coverage of oxygenated VOCs, and integrating satellite-based land use data to refine both anthropogenic and biogenic emission representations. These improvements are essential for advancing air quality forecasting and informing targeted ozone mitigation strategies.

**Acknowledgments**

Special thanks to the Ministry of Environment, Taiwan (MOENV), for providing valuable hourly air quality and PAMS data. This research is funded by the National Science and Technology Council (NSTC), under the contract NSTC 111-2111-M-033-001-MY3, and by the MOENV, under the contracts 111-EPA-F003, MOENV113F018, 111A152, 112AA213, and 113AA083.





**Code/Data availability**

The CMAQ-PAMS model is based on the U.S. EPA Community Multiscale Air Quality (CMAQ) model version 5.2, which
485 is publicly available at https://github.com/USEPA/CMAQ. The modified source codes and scripts used for implementing the
54 PAMS species are available from the corresponding author upon reasonable request. Hourly PAMS observations and air
quality monitoring data are provided by the Ministry of Environment, Taiwan (MOENV). These datasets are publicly
accessible through the MOENV data service portal (https://data.moenv.gov.tw/). Model input data (emission inventories,
meteorology, and boundary conditions) and simulation outputs are available upon reasonable request to the corresponding
author.

**Author contribution**

S.-P. Chen designed, supervised the entire study, and led model development of this research. C.-H. Wang processed the
QAQC of the PAMS observational data. Y.-Y. Lee and F.-Y. Cheng assisted with model configuration and supported
meteorological modeling. J.-L. Wang contributed to the interpretation of results. All authors discussed the findings of the
manuscript.

**Competing interests**

The authors declare that they have no conflict of interest.

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
