# Peer review of "PAMS-Constrained Top-Down Calibration of VOC-Speciated CMAQ Simulations"

_EGUsphere, 2025_

## Author Comment (AC1)

**RC1**: ['Comment on egusphere-2025-4664'](), Anonymous Referee #1, 30 Oct 2025

This manuscript by Chen et al. modified the chemical mechanism of the CMAQ model to incorporate 54 VOC species included in PAMS. Furthermore, the anthropogenic VOC emissions in the emission inventory were adjusted under the constraint of PAMS observation data from Taiwan, resulting in modeled VOC concentration results that are close to the observed levels. This study conducts model evaluation for individual PAMS VOC species.

Main comments:

1. The authors appear to frame the inclusion of PAMS species as a key feature of this study. Based on current research in this field, lumped mechanisms are widely recognized as a reliable method for improving model operational efficiency— specifically by grouping species with similar photochemical reaction behaviors. Different gas-phase chemical mechanisms may treat certain species (deemed important) as individual components, whether in the context of emissions or chemical reaction mechanisms. More notably, the explicit Master Chemical Mechanism (MCM) includes no fewer than 6,000 explicit species. Therefore, the novelty of developing the CMAQ-PAMS modeling system needs to be discussed in Introduction and Discussion section.

Reply: We thank the reviewer for this insightful comment. We agree that lumped checmical mechanisms have long been the standard approach for regional and operational air quality modeling due to their computational efficiency and their abaility to represent the aggregate reactivity of VOC groups. However, despite their widespread use, lumped VOC species cannot be directly compared with observational VOC datasets (e.g., NMHC or PAMS measurements), because only a limited subset of modeled surrogates corresponds to measureabe compounds. As a result, validation of VOC simulations has remained an unresolved issue: most previou stidies evaluate model performance for $O_3$ or $NO_x$ prior to conducting sensitvity or scenario analyses, yet rarely validate VOC simulations due to this incompatibility. This gap maks it difficult to assess whether simulated VOC composition and reactivity are reasonable, even though these precursors critically determine $O_3$ formation behavior.

Although the explicit mechanisms such as the Master Chemical Mechanism (MCM) can represent thousands of VOC species, their computational cost make them impractical for 3-D regional simulations, especially for regulatory or episodic modeling. Conversely, traditional lumped mechanisms (e.g., CB6, SAPRC, RADM2, etc.) provide only a limited number of explicit VOC species-such as isoprene or

several small carbonyl compounds-that can be compared with observations. However, the vast majority of VOCs are represented as lumped surrogates, preventing species-resolved evaluation against comprehensive PAMS measurements.

The novelty of CMAQ-PAMS lies in bridging this long-standing gas between lumped-mechanism modeling and species-resolved VOC observations. In this study, we introduce 54 PAMS-targeted VOC species into a regional CTM without resorting to a fully explicit mechanism, thereby maintaining the computational efficiency and chemical structure of the CB6 framework. At the same time, the model incorporates species-specific emission mapping and output configurations that allow direct comparison with PAMS measurements. This development establishes a new evaluation framework in which modeled and observed VOCs can be compared at the individual-species level—an outcome that has not been achieve using lumped mechanisms alone. As a result, CMAQ-PAMS substantially improves diagnostic capability for source attribution, emission speciation assessment, and photochemical regime classification (e.g., VOC-limited vs. $NO_x$-limited conditions), now grounded in PAMS-resolved VOC reactivity rather than aggregated surrogates.

These perspectives are now explicitly discussed in the manuscript (Introduction: lines 43-49, 87-109; Results and Discussion: lines 305-310, 386-393, 436-440, and 551-557). The CMAQ-PAMS does not aim to replace established lumped mechanisms not to emulate a fully explicit mechanism like MCM. Rather, its contribution is an intermediate and operationally feasible system that provides species-resolved VOC simulation specifically aligned with regulatory PAMS observations, improving both model evaluation and chemical diagnosis for regions where speciated VOC data are available.

2. Although Figure 1 is provided to introduce the CMAQ-PAMS modeling system, it only shows where modifications were made to the model. The process of inventory adjustment and model modification are needed. For instance, how to generate emissions of PAMS species into the CB mechanism via SPECIATE. Mabe it is necessary to include at least one table in the appendices that presents the proportion of each PAMS species within lumped species, preferably disaggregated by emission source.

Reply: We appreciate the reviewer's constructive suggestion. In addition to revising Figure 1 to explicitly indicate where CMAQ-PAMS modifies both the emissions processing and the gas-phase chemical mechanism, we have expanded the model Section 2.1.3 to include a detailed explanation of the inventory-adjustment procedure (in lines 161-165). The revised text now describes how anthropogenic VOC emissions were disaggregated into 54 PAMS species using source-specific

SPECIATE profiles and how these PAMS species were subsequently integrated into their CB6 lumped VOC groups for chemical processing (in lines 130-135).

To further enhance transparency, we have added a new table to the Supplementary Information (Table S3), which presents the proportional contribution of each PAMS species within its respective CB6 lumped group, based on domain-wide anthropogenic emissions. This table documents the final speciation structure used by the CMAQ-PAMS emission processor and provides a clear quantitative link between the SPECIATE-based disaggregation and the implementation of the CB6 mechanism. We believe these additions fully address the reviewer's request for clarification regarding the inventory adjustment process and the model modifications supporting CMAQ-PAMS.

3. There is no doubt that using VOC observation data to constrain emissions is an effective method for improving VOC simulation performance. However, details are needed for the approach used to adjust emissions:

- The manuscript states that VOC observations from 12 sites were used to modify emissions based on the ratio of simulated to observed values. Were the remaining grid points calculated via linear interpolation (Lines 314–316)?

- Was this adjustment applied to the total emissions for the study period or conducted hourly?

- When referring to "the total VOC amount is conserved," does this mean the total VOC emissions in the inventory remained unchanged (and if so, were the proportions of other VOC species reduced?)? Or does it mean emissions of PAMS-corresponding VOCs remained unchanged?

- It is essential to include the calculation formula for this emission adjustment process to clarify these ambiguities.

Reply: We thank the reviewer for pointing this out. The previous wording has been corrected because the total VOC mass is "not" conserved in our adjustment procedure. Consistent with Chen et al. (2014), the PAMS-species emissions were scaled according to the simulated-to-observed ratios, and the resulting increase or decrease in PAMS emissions was directly incorporated into the inventory, thereby modifying the total VOC amount. No compensatory reduction was applied to non-PAMS species. Asa result, the adjusted emissions alter the total VOC mass, rather than redistributing it among existing species. This clarification has been added to the revised manuscript in a new Section 2.1.4.

4. Lines 90–91: Adjustments to PAMS emissions would likely have a significant impact on $O_3$ formation—insights that could help us better understand the role of VOC emissions in pollutant simulation. What are the results of O3?

Reply: We thank the reviewer for raising this insightful comment. In this study, the adjustment of PAMS emissions serves as a necessary preprocessing step to obtain a chemically consistent VOC inventory that can be directly compared with PAMS observations. Because the revised VOC emissions are used as the baseline configuration for all subsequent simulation, the $O_3$ results presented in Section 3.5 already reflect the influence of the PAMS-adjusted emissions.

Accordingly, the manuscript does not include a separate comparison of $O_3$ concentrations before and after the emission adjustment. Instead, all $O_3$ analyses in the Results and Discussion section correspond to simulations that incorporate the updated PAMS emission speciation. This clarification has been added to the revised manuscript (Lines 394-396).

Minor comments:

1. There are multiple instances of inconsistent subscript formatting for $O_3$ and $NO_x$ throughout the manuscript, such as in Line 9, Line 29, Line 35, and Line 37. Please standardize subscript formats; as inconsistencies may be interpreted as evidence of AI-generated content.

Reply: Thank you for pointing out the inconsistent subscript formatting in the chemical species (e.g., $O_3$ and $NO_x$). We have carefully reviewed the entire manuscript and standardized all chemical formulas to use proper subscript formatting. The inconsistencies now in Lines 10, 30, 32, and 43—as well as elsewhere in the text—have now been corrected. We appreciate this comment, as consistent notation improves readability and avoids any confusion regarding formatting quality.

2. Lines 34–39: This section summarizes the research status of VOCs in air quality models, but every sentence lacks supporting literature citations.

Reply: Thank you for this valuable comment. We agree that the discussion on VOC representation in air quality models should be supported with appropriate literature. In the revised manuscript, we have added citations to foundational and recent studies that describe (1) the chemical diversity of VOCs, (2) challenges in emission estimation and model evaluation, and (3) the use of lumped chemical

mechanisms in regional CTMs. These citations help strengthen the context and grounding of our statements regarding current research status.

3. There are numerous issues with the reference citation format in the manuscript. For example: Line 50: Cardelino and Chameides (Cardelino and Chameides, 1990)→Cardelino and Chameides (1990); Line 87: (Knote et al.) → (Knote et al., 2015). Please standardize all reference citations to comply with the journal's formatting requirements.

Reply: Thank you for this helpful comment. We have thoroughly reviewed all in-text citations throughout the manuscript and corrected inconsistencies to ensure full compliance with the journal's reference formatting requirements. Specifically, issues such as duplicated author names (e.g., "Cardelino and Chameides (Cardelino and Chameides, 1990)") and misleading of citation (e.g., "(Knote et al.)") have been corrected to the appropriate forms (e.g., "Cardelino and Chameides (1990)" and "hydroxyl (OH)"). We also conducted a comprehensive check to ensure that all citations follow a consistent and standardized format.

4. Sections 2–4 can be merged into a single chapter titled "Materials and Methods".

Reply: Thank you for the suggestion. We agree that merging Sections 2–4 into a single chapter titled "Materials and Methods" improves clarity and aligns with common journal structures. In the revised manuscript, we have combined these sections accordingly and reorganized the subsections to ensure a logical flow and avoid redundancy. This restructuring enhances readability and better reflects the methodological framework of the study.

5. Lines 142–144: For the 2021 simulation, the 2010 emission inventory used in the manuscript appears outdated. Although the simulation performance of $NO_x$ and $O_3$ shown in Figure S3 seems acceptable, statistical parameters (e.g., $R^2$, NMB, NME) should be added to verify the simulation accuracy.

Reply: Thank you for the comment. The description of the emission inventory has now been clarified to avoid the misunderstanding that a 2010 inventory was directly applied to the 2021 simulation. In our modeling system, The East Asia boundary/parent domain (D1) uses the MICS-Asia 2010 inventory with the officially recommended projections of China's anthropogenic emissions to the year 2017, following Li et al. (2017) and Zheng et al. (2018), which is the most recent large-scale dataset available for regional boundary conditions.

For the Taiwan domain (D2), we use the Taiwan Emission Database System (TEDS) version 11.0, whose baseline year is 2019, and which is the official national

emissions inventory adopted by Taiwan's Ministry of Environment. TEDS11 is considered representative for the 2021 baseline scenario, given the absence of major structural emission changes between 2019 and 2021.

Additionally, we agree that quantitative model evaluation is important. Therefore, $R^2$, NMB, and NME values for $NO_x$, and $O_3$ have been added (Table S4), confirming that the model performance meets widely used regional documents.

6. Improve the standardization of figures and tables. This encompasses, but is not limited to, Figure 3, Table 3, Figure 5, Figure 6, Figure 8, and most figures in the supplementary information. Specifically: Use a consistent Arial font across all figures; Standardize axis scales and labels; Ensure no single figure spans multiple pages. These adjustments are necessary to meet the journal's formatting standards.

Reply: Thank you for the valuable suggestions regarding figure and table standardization. We have carefully revised all figures and tables in both the main manuscript and the supplementary information to comply with the journal's formatting requirements. Specially for the consistent font usage (using Arial), standardized axis scales and labels (harmonized across comparable figures), and figure layout adjustments (prevent any single figure from spanning multiple pages).

7. Lines 350–356: Does Table 2 present the average values of adjustment coefficients? If so, this should be clearly indicated in the table title to avoid ambiguity for readers.

Reply: Thank you for the helpful suggestion. Yes, Table 2 presents the average adjustment coefficients derived for each PAMS species. To ensure clarity for readers, we have revised the table title to explicitly state that the values shown represent average emission adjustment factors.

8. Line 375: The manuscript mentions that the improved simulation includes explicit ISOP compared to previous mechanisms. However, many current research already considered and analyzed ISOP as an individual species. The authors should clarify: What distinguishes your results from these prior studies?

Reply: Thank you for this important comment. We agree that isoprene has been explicitly represented in many contemporary chemical mechanisms and has been extensively analyzed in prior literature (e.g., Guenther et al., 2012; Palmer et al., 2006; Steiner & Goldstein, 2007; Curci et al., 2010). In our study, the key distinction is that isoprene emissions are overwhelmingly dominated by biogenic sources, and therefore no anthropogenic emission adjustment factors were applied to ISOP. Instead, ISOP emissions were directly taken from the MEGAN biogenic emission model without modification.

The contribution of our work is that **it represents the first island-wide, station-resolved temporal comparison of simulated and observed isoprene concentrations in Taiwan.** While previous studies have discussed isoprene chemistry or biogenic emission modeling more broadly, detailed **time-series evaluation against in situ PAMS observations across multiple stations in Taiwan has not been documented in prior research**. Therefore, our analysis fills a critical gap by providing the first comprehensive observational evaluation of isoprene simulation performance in a Taiwan-focused CMAQ-PAMS modeling framework.

To address your concern, we have clarified the novelty of our work by adding explanatory text in Lines 480–484. The revised manuscript now includes the following sentence: *"Although isoprene has been explicitly represented and analyzed in many prior studies (e.g., Guenther et al., 2012; Palmer et al., 2006), in Taiwan its emissions are overwhelmingly biogenic. Therefore, no emission adjustment factors were applied to ISOP in this study. Instead, our contribution lies in providing the first island-wide temporal comparison between simulated and observed isoprene concentrations across PAMS stations, offering new insight into the model's ability to capture biogenic emission-driven variability."*

**RC2**: 'Comment on egusphere-2025-4664', Anonymous Referee #2, 05 Nov 2025
General Comments

In this study, the authors have incorporated speciated VOC measurements from Taiwan's PAMS network into the CMAQ model, addressing the limitations of conventional lumped-species chemical mechanisms. The revised model is now capable of explicitly simulating 54 PAMS-targeted species, more direct and comparable evaluation against observed VOC data. This work is timely and valuable, however, there exists some issues which need to be addressed before publishing. Therefore, it requires substantial revisions to improve its clarity and presentation. I recommend a major revision. The specific suggestions are as follows:

Reply: We sincerely thank the reviewer for the careful evaluation of our manuscript and for recognizing the significance of integrating Taiwan's PAMS speciated VOC measurements into CMAQ to enable explicit simulation of 54 VOC species. We appreciate the reviewer's constructive feedback and acknowledge the need for substantial revisions to improve clarity, organization, and presentation. In response, we have thoroughly revised the manuscript and addressed all specific comments point-by-point. We believe these revisions have strengthened the scientific rigor and readability of the work. Detailed responses to each comment are provided below.

Specific Comments

1. Lines 25–30: Could the authors clarify which substances are included under the term "most primary pollutants"?

Reply: Thank you for the comment. To avoid ambiguity, we have clarified which substances are included under the term *"most primary pollutants."* In the revised manuscript, we now explicitly state that this refers to pollutants such as $NO_x$, CO, VOCs, all of which have exhibited long-term declining trends in Taiwan due to regulatory controls. The text has been rephrased as in Line 29: "… most primary pollutants, including $NO_x$, CO, VOCs, have shown a declining trend (Chen et al., 2021)."

2. Lines 35–40: While "grouping chemically similar VOCs into lumped surrogate species" is a common approach, it is not the only method used. I suggest the authors provide a more comprehensive overview of chemical mechanisms employed in chemical transport models (CTMs).

Reply: Thank you for your constructive suggestion. We agree that the original text overly emphasized lumped-structure mechanisms. In the revised manuscript, we have expanded the overview of chemical mechanisms used in CTMs to provide a

more balanced and comprehensive description. Specifically, we now distinguish between lumped surrogate mechanisms (e.g., CB05, CB6, SAPRC) and explicit or semi-explicit mechanisms (e.g., MOZART, RACM2, MCM), noting their respective advantages and computational trade-offs. This broader context helps clarify how different mechanisms represent VOC species in regional air quality modeling.

The text has been rewritten as in Line 37-41: "Many widely used mechanisms, such as the Carbon Bond families (Yarwood et al., 2005) and SAPRC families (Carter, 2010), reduce computational cost by grouping chemically similar VOCs into lumped surrogate species. Other mechanisms, including RACM2 (Goliff et al., 2013), MOZART (Emmons et al., 2010), and the near-explicit Master Chemical Mechanism (MCM) (Metzger et al., 2008), provide more detailed or explicit representations of VOC oxidation pathways, albeit with substantially greater computational demands."

3. Lines 70–75: The Introduction lacks sufficient context regarding the significance and necessity of developing CMAQ-PAMS.

Reply: Thank you for this insightful comment. We agree that additional context is needed to better motivate the development of CMAQ-PAMS. In the revised manuscript, we have expanded the Introduction to clarify the scientific and practical significance of CMAQ-PAMS. Specifically, we now highlight: (1) the lack of speciated VOC representation in conventional CMAQ simulations, (2) the importance of reproducing individual PAMS species for process-level understanding of ozone formation, and (3) the need for improved tools to reconcile discrepancies between model simulations and PAMS observations. These additions provide clearer justification for developing and applying CMAQ-PAMS in this study.

The text in Lines 80-83 are revised as "… *However, conventional CMAQ configurations do not explicitly resolve the full suite of PAMS species, limiting their ability to diagnose species-specific contributions to ozone formation or to evaluate model performance against detailed observational datasets….*", and in Line 102-109 as "*This gap is critical because individual VOCs exhibit highly variable reactivities, emission sources, and sensitivities to control strategies. Developing CMAQ-PAMS—an enhanced version of CMAQ capable of simulating individual PAMS species—provides a necessary framework for directly linking model outputs with observational constraints, refining VOC emission inventories, and improving mechanistic understanding of ozone formation processes. By enabling species-level comparisons between simulations and PAMS observations, CMAQ-PAMS facilitates a more rigorous assessment of VOC model performance than has been possible with standard lumped-species mechanisms. This capability is particularly important for regions like Taiwan, where ozone episodes are strongly influenced by*

*speciated VOC chemistry but where emission inventories and model representations remain uncertain.*"

4. Lines 80–85: Please provide detailed information on the 54 PAMS species and their corresponding mappings in the CB05e51 and CB6r3 mechanisms directly.

Reply: We thank the reviewer for this helpful suggestion. To address the comment, we have added a clear description in Lines 82–86 explaining how each of the 54 PAMS species is mapped onto the corresponding lumped VOC species in the CB05e51 and CB6r3 mechanisms. Because the full mapping table is lengthy, the complete list—containing all PAMS species, their chemical structure classification, and their associated CB05e51 and CB6r3 surrogate species—is now included in the Supplementary Information as Table S1. This provides the requested level of detail and allows readers to directly examine how individual PAMS species are treated within both chemical mechanisms.

5. Lines 85–90: The reference to (Knote et al.) appears to be incomplete.

Reply: Thank you for pointing out this issue. The reference "(Knote et al.)" now in Lines 123-126 was indeed incorrect and was intended to denote the hydroxyl radical instead of a literature citation. In the revised manuscript, we have replaced "(Knote et al.)" with "(OH)" in Line 125.

6. line100: The additional chemical reaction pathways should be presented as supplementary.

Reply: We thank the reviewer for this helpful suggestion. Additional reaction pathways associated with the newly introduced PAMS species have now been compiled and included in the Supplementary Information (Table S2), as well as additional text content in Section 2.1.1 (Lines 136-138). Although several reaction concepts were adapted from our earlier PAMS-AQM development (Chen et al., 2010), all pathways have been reformulated to ensure full compatibility with the CB6 chemical mechanism used in CMAQ-PAMS. Only the reactions that were newly added or modified in this study are listed, enabling readers to reproduce the CMAQ-PAMS chemical configuration without redundancy.

7. lines 119-121: The processes of converting VOC concentrations into emission rates and spatially allocating these emissions within a gridded inventory remain unclear.

Reply: We appreciate the reviewer's comment and agree that the description in the manuscript was not sufficiently clear. In this study, VOC concentrations were not converted directly into emission rates. Instead, following the top-down adjustment

approach of Chen et al. (2014), the observed-to-simulated PAMS VOC ratios were used to derive hourly species-specific correction factors. These factors were then applied to the original PAMS emissions to scale the emission upward or downward. Thus, the spatial allocation of emission remains identical to the original TEDS inventory, and only the species-level magnitudes are adjusted. The revised manuscript now clarifies this process (in Section 2.1.4) and explicitly states that total VOC mass is not conserved after applying the adjustments.

8. Lines 145–150: The statement "the transboundary influence for the PAMS species is limited" represents a rather bold assumption. What evidence supports this claim? It also appears to be inconsistent with the results discussed in Section 5.3.

Reply: We appreciate the reviewer's insightful comment. We have clarified the justification for the assumption directly based on the evidence within the manuscript and have revised the text to avoid any ambiguity.

Our statement that "the transboundary influence for the PAMS species is limited" is based on the chemical characteristics and observations of the 54 PAMS species during the September-October 2021 case. As described in the manuscript:

1. Most PAMS species are short-lived primary NMVOCs: These species-primary C2-C9 alkanes/alkenes/aromatics-react rapidly with OH, $O_3$, and $NO_3$ radicals (Section 2.1), and therefore cannot sustain long-range transport across the East Asian continent with lifetimes of only minutes to several hours. This is the reason PAMS species are modeled with loss processes only and do not contribute to boundary conditions (lines 223-228).
2. Even the relatively long-lived PAMS alkanes show minimal contribution during LRT events: As described in Section 2.2.1, a small subset of PAMS species (ethane, propane, butanes) have longer lifetimes and can be detectable in long-range transported air masses (Chang et al., 2022). However, the revised text also clarifies that these long-lived alkanes exhibit only minimal mixing ratios in LRT air compared with local concentrations due to extensive dilution over the ocean. Thus, even for the longest-lived species, the transported signal is negligible relative to local emissions, consistent with our assumption.
3. Observations during the 2021 event show no transboundary enhancement of PAMS species: Synoptic patterns (Fig. S1) show that although East Asian circulation affected $O_3$, PAMS species did not exhibit coherent large-scale enhancements that are characteristic of long-range pollutant arrival. Also, time series at all PAMS sites (Fig. S3-S4) show strongly localized patterns with site-specific diurnal signals, rather than regional spikes expected from transported plumes. These figures demonstrate that while $O_3$ and secondary pollutants can

be transported, primary PAMS VOCs do not display a regional transport signature.

4. Section 5.3 (now in Section 3.3) does not discuss $O_3$ or transboundary influence; therefore, there is no inconsistency: The reviewer raised concern about inconsistency with Section 5.3. However, Section 5.3 ("In-depth diagnosis of ModSIM by CMAQ-PAMS", now in Section 3.3) focuses exclusively on local emission characteristics of toluene and isoprene-including their emission distributions, diurnal cycles, and interactions with local meteorology. This section does not discuss: transboundary transport, regional inflow, or ozone formation. Instead, it strictly evaluates how adjusted emissions improve local VOC simulations.

To improve clarity, we revised the original sentence in Line 223-228: "... *Because the 54 PAMS species are short-lived primary VOCs, their direct long-range transport from outside Taiwan is expected to be minimal. This assumption applies only to PAMS species themselves and does not affect discussions related to ozone or secondary products in later sections*."

9. Line 164, figure S2: Is it showing the O3 vertical profile? What does the left axis stand for? How were the vertical winds observed?

Reply: We thank the reviewer for the question and we clarify that Figure S2 does not show $O_3$ vertical profiles. The figure shows surface $O_3$ time series plotted as a function of time and station index, along with surface horizontal wind vectors from Taiwan's air quality monitoring network. To avoid misunderstanding, we revise the caption in Figure S2 as "Fig. S2. Hourly observations of $O_3$ and horizontal surface wind vectors (10m) at all AQS from north to south of Taiwan (IS→NT→CM→CT→YCN→KP→YHD) for the selected case in 2021 (2021/09/27-10/03). The y-axis indicates station sequence, not altitude."

10. Section 3.3: What is the rationale for comparing the computational times of the new (cb6rpams_ae6) and the old (cb6r3_ae6) chemical mechanisms in this study?

Reply: We thank the reviewer for the question. The comparison of computational times between the original CB6 (cb6r3_ae6) mechanism and the new PAMS-extended mechanism (cb6r3pams_ae6) was included for the following reasons:

1. The PAMS mechanism substantially expands the chemical system, and the added computational burden must be documented, such as the new CMAQ-PAMS mechanism explicitly simulated 54 additional VOC species, increasing the number of model-integrated gas-phase species from 219 species (cb6r3_ae6) to 275 species (cb6r3pams_ae6), as shown in Table 1. Also the

number of chemical reactions, the dimensionality of the ODE system solved by the EBI solver, memory usage and output file sizes.

2. CMAQ-PAMS is intended to be used to operationally and in future research; model users need to know the performance impact: How much additional wall-clock time is required, how CPU usage scales with the enlarged chemical mechanism, and whether the computational overhead remains manageable.

3. Including performance metrics is standard practice when introducing a modified CTM mechanism: when a new chemical mechanism, solver configuration, or model module is introduced, performance benchmarking is commonly reported in the atmospheric modeling literature.

To address the reviewer's concern, we have added the following explanation at the beginning of Section 2.3.3: "*Because CMAQ-PAMS introduces 54 additional VOC species and substantially increases the size of the gas-phase chemical system, it is necessary to quantify the computational overhead relative to the original CB6 mechanism. This comparison allows model users and operational centers to assess the feasibility and resource requirements of adopting the expanded mechanism.*"

11. It is strongly recommended that Sections 2, 3, and 4 be combined into a single "Materials and Methods" section to enhance the overall readability of the paper.

Reply: We appreciate the reviewer's helpful suggestion. In the revised manuscript, Sections 2, 3, and 4 have been consolidated into a single unified "Materials and Methods" section.

12. Figure 3. Why are only the results for the W-site presented? How do the results from other types of sites compare, and what are the key similarities or differences among them?

Reply: Thank you for the insightful comment. The W-site (Wanhua) was selected as the representative example in Figure 3 because it clearly illustrates the four typical simulation behaviors observed across the PAMS species-namely comparable, overestimated, underestimated, and missing (zero) emissions. Results from all other PAMS stations are already included in the Supporting Information (Figure S4).

As for the key similarities across all PAMS stations, OrigSIM shows substantial species-specific biases (over/underestimation or missing species), reflecting the limitations of U.S.-based VOC speciation profiling. ModSIM significantly improves agreement with observations after top-down calibration, reducing biases and recovering missing species. Diurnal patterns are generally well captured at all stations, indicating that physical transport and photochemistry are adequately represented. Temporal alignment of peaks is consistent across stations, showing that

meteorological fields are properly simulated. These similarities demonstrate the CMAQ-PAMS framework is spatially robust across very different environments.

Although the four simulation patterns appear at all sites, some regional characteristics are noted: (1) Urban sites (W, Z1, P1, T3) tend to show stronger signals from traffic and solvent-use VOCs (e.g., toluene, xylenes); ModSIM corrects underestimations for these species. (2) Industrial or mixed sites (T1, P2) often exhibit sharper peaks related to local emission variability; ModSIM still captures the amplitude better than OrigSIM. (3) Southern sites (T3, X, C) show slightly higher nighttime VOC accumulation under weaker boundary-layer mixing; ModSIM also improves the magnitude and pattern. (4) Coastal or rural sites (T2, Z2, Q, D) have lower VOC abundance overall, with diurnal patterns driven by advection and boundary-layer evolution; ModSIM reduces biases while maintaining the correct temporal shape.

Despite these local differences, the overall conclusions remain consistent: ModSIM exhibits major improvements across all conditions, and the W-site example effectively captures the general model behavior.

To improve clarity, we added the following statement in Section 4.1.1 in Line 295-297: "... *The W-site is shown as a representative example because it exhibits all four characteristic modeling behaviors observed across the PAMS stations. Full results for all 12 stations, demonstrating similar patterns, are provided in Figure S4....*"

13. Lines 365–370: What exactly is meant by "local circulation" in this manuscript? In what ways does it affect the spatial distribution of pollutants?

Reply: We thank the reviewer for the opportunity to clarify this term. In the manuscript, "local circulation" refers to the thermally and topographically driven wind patterns within Taiwan, including land-sea breeze circulations, mountain-valley winds and terrain-channeled flows along the western plains. These circulations dominate surface wind fields during the selected 2021 episode, which was characterized by weak synoptic forcing and stagnant conditions (Section 2.2.2; see also Fig. S1).

To avoid ambiguity, we have revised the text in Lines 467-470 as follows: "... *The spatial distribution of toluene is strongly influenced by local circulation, referring to diurnally driven land-sea breezes and terrain-channeled valley winds that dominate surface flow under weak synoptic conditions. These circulations transport emissions downwind and reshape concentration patterns, resulting in spatial maxima that do not always coincide with the emission hotspots....*"

14. Figure 6: The title, figure number, content, and caption should remain together rather than separated. Furthermore, to enable a more straightforward comparison, the units (kg/hr and ppbC) should be converted and the time zones harmonized.

Reply: We thank the reviewer for the helpful suggestions. Both formatting and scientific clarity have been improved in the revised manuscript: In the revised version, Figure 6, its panels, and its caption are placed together on the same page. Units are intentionally different because the figure compared emissions vs. concentrations, but we have clarified this in the revised caption. Time zones have now been harmonized.

We have revised the caption to explicitly state: "Figure 6: … Panels (a) and (c) show emission fluxes (kg/hr), whereas panels (b) and (d) show simulated ambient mixing ratios (ppbC). These quantities represent different physical variables and are therefore not directly convertible. All times are shown in CST (UTC+8), consistent with PAMS and TAQMN observations."

15. It is recommended to provide relevant evidence to demonstrate that Taiwan is in a "VOC-limited regime".

Reply: We thank the reviewer for raising this important statement. The manuscript has been revised to provide clearer evidence that the study period and major urban areas in western Taiwan are predominantly VOC-limited. The supporting evidence is based on citations, consistent with established ozone chemistry diagnostics.

We added the following explanatory sentence in Section 3.5 (in Lines 551-557): "*Numerous field and modeling studies have demonstrated that major urban and industrial regions in western Taiwan frequently operate under VOC-limited ozone formation regimes. Early observational analyses showed that ozone formation in Taipei and northern Taiwan is constrained by VOC availability (Chang and Lee, 2006; Wu et al., 2006). Similar VOC-sensitive behaviour has been reported in central Taiwan (Shiu et al., 2007). More recent analyses further confirm this pattern: long-term NOx reductions have led to higher ozone levels across Taiwan (Chen et al., 2021), and scenario-based modeling shows that NOx-only reductions increase ozone in northern Taiwan, consistent with a VOC-limited regime (Chuang et al., 2022). Collectively, these independent studies strongly support the VOC-limited interpretation applied in this study.*"

Technical Corrections

1. The citations in the manuscript are not properly formatted. For example, in lines 50-60, "Yang et al. (Yang et al., 2005) analyzed …" should be "Yang et al. (2005) analyzed …". In lines 70-75, "(Chen et al., 2010; Ying and Li, 2011; Chen et al., 2014a; Chen et al.,

2015; Su et al., 2016). (Ge et al., 2024; Rowlinson et al., 2024)" should be "(Chen et al., 2010, 2014a, 2015; Ge et al., 2024; Rowlinson et al., 2024; Su et al., 2016; Ying and Li, 2011)".

Reply: We thank the reviewer for pointing out the inconsistences in the citation formatting. All in-text citations have now been carefully checked and corrected throughout the entire manuscript to follow a consistent and journal-compliant style.

2. The manuscript uses both "ozone" and "O3" in different places; it is recommended to use a consistent notation throughout.

Reply: We appreciate the reviewer's helpful suggestion. In the reviewed manuscript, we have standardized the terminology by using "$O_3$" consistently throughout the text, figures, tables, and captions. The term "ozone" is now only used when referring to aerosol or gas species in a general narrative context (e.g., "ozone pollution," "ozone episodes"), while all scientific or quantitative references (e.g., concentrations, production rates, chemical mechanisms) use the symbol "$O_3$" for clarity and consistency.

3. Lines 95: Change to "Figure 1. The CMAQ modeling framework revised for CMAQ-PAMS (red parts)."

Reply: We thank the reviewer for the suggestion. The caption for Figure 1 has been revised accordingly.

4. A few spelling and grammatical errors in the manuscript need to be corrected.

Reply: Thank you for pointing this out. The entire manuscript has been carefully proofread, and all spelling and grammatical errors have been corrected. In addition to standard corrections, we also improved sentence flow and consistency, especially in the Introduction, Methods, and Discussion sections. We appreciate the reviewer's suggestion, which has helped enhance the readability and clarity of the manuscript.